# *AGENTS.md* for Reconstructing a Tiny GenAI Library: Initial Analysis of Trivial PyPI Packages

## Abstract

The rise of agentic systems offers a dynamic alternative to traditional third-party library reuse, promising on-the-fly functionality adaptation. Central to this shift is AGENTS.md, which serves as the contextual guide for navigating Generative Artificial Intelligence (GenAI) environments. In this new ideas paper, we explore how libraries could be generated on-demand through deconstructing the specification contexts in AGENTS.md. Our study involves generating 15 different combinations of AGENTS.md to generate 5,760 specifications and 80 GenAI libraries from 40 randomly sampled libraries. Our initial analysis reveals that context is critical: function descriptions and implementations account for the most of generated documentation, while testing and overviews generate the least. With the growth of AI Slop, we empirically show that specifications have potential to guide GenAI for tiny code compared to human-written equivalents. We finally present a research agenda for efficient, on-demand GenAI library technologies.

## Keywords

AGENTS.md, Empirical Study, Generative AI, Software Libraries

**ACM Reference Format:**
Anonymous Author(s). 2026. *AGENTS.md* for Reconstructing a Tiny GenAI Library: Initial Analysis of Trivial PyPI Packages. In *Proceedings of Make sure to enter the correct conference title from your rights confirmation email (Conference acronym 'XX)*. ACM, New York, NY, USA, 5 pages. https://doi.org/XXXXXXX.XXXXXXX

## 1 Introduction

Agentic coding for Software Engineering has led to developers using human-written natural language objectives that are autonomously planned and executed with minimal human involvement [4]. Despite its capabilities, the autonomy of Artificial Intelligence (AI) agents remains dependent on a critical factor generated by humans: *context* [3]. Suri et al. demonstrated that when dealing with rather complex code generation, specific context becomes necessary to guide the AI agent [12]. Although they experimented using concise, high-level prompts, this led the AI agent to conflicts and ambiguity when the prompt did not have detailed context. This implies the need for accurate context within the prompt for agentic coding. To improve the AI agent's understanding of the developer's intent, agent context files such as AGENTS.md have been implemented

in software projects [9]. These files act as READMEs for AI agents, providing context and instructions to assist developers [2]. They contain various types of information, such as coding conventions, build commands, and the purpose of the project, which serve as specifications for the software project [10].

Beyond the emergence of agentic coding, software development still heavily relies on open-source libraries with reusable functionality, significantly reducing development time and effort compared to building functionality from scratch [8]. Among these countless software libraries are small, minimalist libraries that accommodate simple and trivial tasks. Previous studies have referred to these types of libraries as trivial [1] or micro-packages [6]. To better understand whether a library is considered trivial, prior studies have introduced constraints regarding its lines of code and cyclomatic complexity.

In this new ideas paper, we investigate the relationship between specifications and GenAI by generating trivial libraries, as they have relatively small and simple specifications. Based on prior work [1], we define a Python trivial library as one that satisfies one of the following conditions: a total of $\leq 35$ lines of file, or a maximum value of McCabe's Cyclomatic Complexity $\leq 10$ in a single function. We use a variety of contexts as specifications for generating these trivial libraries. Specifically, we explore (1) which types of context generate the *most* and *least* specification in terms of lines of file, and (2) how the GenAI-generated library compares to the original human-written library. In detail, we use metrics including Lines of File (LoF), File Count (FC), McCabe's Cyclomatic Complexity (CC), and the number of generated functions.

We report two findings. First, the context of *Function Description* and *Implementation Detail* generates the greatest amount of specification, while the context of *Repository Overview* and *Testing* generates the least amount of specification. The second finding is AI-generated libraries tend to have *fewer LoF* and *lower FC*, regardless of the amount of specification provided. AI-generated libraries exhibit a higher maximum CC within a single function.

Future work includes a more comprehensive study on the specificationa and detailed function-level validation of the GenAI library against the human-written library. All scripts and data are available at https://anonymous.4open.science/r/acm-aiware-2026-reconstructing-trivial-pypi-packages/ [1]

## 2 Empirical Design

Figure 1 shows our complete library reconstruction pipeline. Our experiment is divided into three parts.

***Data Collection Process.*** The left part of Figure 1 explains our data collection process. We used the PyPI dataset from a previous study [5] to collect trivial library samples, applying statistical sampling with a confidence level of 95% and a 5% margin of error.

---

[1]full dataset will be disclosed after acceptance.

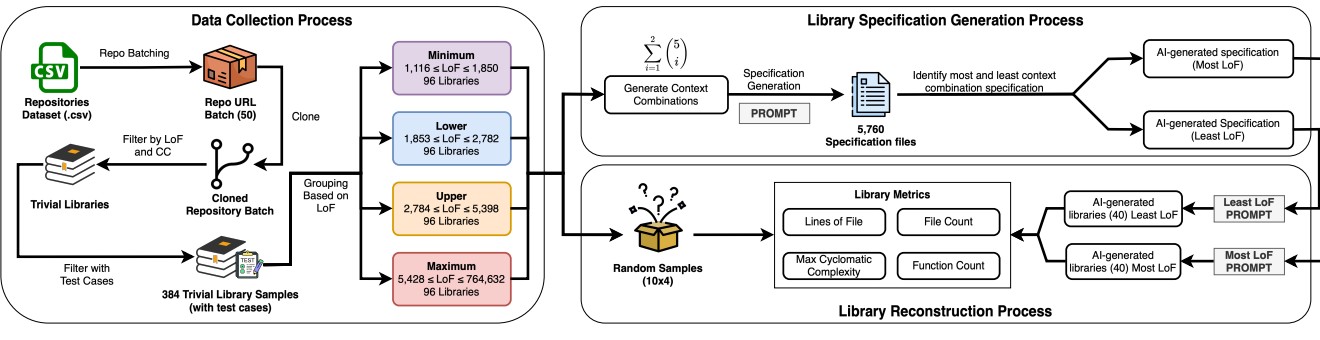

**Figure 1: Overview of experiments that include the data collection, specification and library generation processes**

This yielded 384 trivial library samples from 733,544 PyPI packages available at the time of the study. We used two conditions to collect trivial library samples. The first condition was that the library must have either ≤ 35 LoF in total, or ≤ 10 CC in one function. The second condition was that the library must have at least one test case so that we can verify the AI-generated libraries using the original libraries' test cases in future research. To calculate CC, we utilize Radon [7] to compute the CC of all functions within a library. We then identify the highest CC value among those functions. If this value is ≤ 10, we classify the library as trivial.

Upon gathering the trivial library samples, we divided them into four groups based on their overall LoF using the interquartile range (IQR) of all 384 samples. This resulted in an even distribution of 96 libraries in each group. The purpose of this grouping is to observe the behavior of the AI agent when dealing with both small and large libraries.

***Specification Generation Process***. Instead of using the original documentation created by the libraries' developers, we use AI agents to analyze the trivial libraries and generate AGENTS.md files for each library in markdown format. For our experimental environment, we use Visual Studio Code as the Integrated Development Environment, along with the GitHub Copilot extension and GPT-4.1 as the Large Language Model (LLM). We use the Copilot extension for Visual Studio Code rather than Copilot CLI (Command Line Interface) to avoid hallucinations generated by the AI agent. These hallucinations include improper responses, such as the agent being unable to scan the working directories, failing to generate AGENTS.md files, and exhibiting extended reasoning time without timeout. Alongside manual prompting, we also implement an automated verification process to validate both the input (prompt) and the output of the AI agent. This ensures that the collected AGENTS.md files are well-structured and valid.

To maximize the agent's understanding, we constructed the prompt components based on the model defined by Tafreshipour et al. [13].

> {**Specification Prompt**}
> *You are an autonomous software-documentation agent.*
> *You are tasked with generating documentation for*
> *a code repository. This repository contains a trivial*
> *library, and your goal is to produce a comprehen-*
> *sive documentation file that AI agent could read and*
> *use to recreate the repository, including the code and*

> *its functionality. Create the documentation in a file*
> *named AGENTS.md.*
> *The documentation should include the following:*
> ***{Context}***
>
> *...*
> (Full prompt written in our repository)[2]

For the prompt context, we use five different types of context in the prompt. These contexts were selected based on a previous study on the different kinds of instructions developers include in agent context files [3]. We use the following contexts in the specification generation prompt:

(1) **Repository Overview**: A brief description of the repository's purpose.
(2) **Function Description**: Detailed information about the functions used in the repository.
(3) **Implementation Detail**: A summary of the algorithms, logic, and design patterns used in the repository.
(4) **Testing**: Examples of test cases or verification methods used to validate the repository.
(5) **Reconstruction Steps**: Instructions for recreating the repository from scratch.

To examine whether each context affects the agent's behavior in generating the library specification, we use up to two contexts per prompt to better understand which contexts are necessary to reconstruct a trivial library. As a result, we have a total of $\sum_{i=1}^{2} \binom{5}{i} = 15$ different context combinations. Among these, five are single-context prompts and ten are paired-context prompts. We applied all fifteen context combination prompts to all 384 trivial library samples, resulting in 5,760 AGENTS.md files overall.

Aside from observing the impact of these specification files on the reconstruction process, we also conducted an analysis[2] to determine which context combinations resulted in the most and the least amount of generated specification files. The steps for calculating the most and least extensive specification files are as follows:

- *Step 1*: Iterate through each library.
- *Step 2*: From 15 generated specifications combinations, identify the highest and lowest LoF (i.e., Lines of Files).
- *Step 3*: Tally each combination for each group (i.e., Minimum, Lower, Upper, Maximum).

---

[2] anonymous.4open.science/r/acm-aiware-2026-reconstructing-trivial-pypi-packages

**Table 1: Least Generated LoF Specification (Min=Minimum, Low=Lower, Upp=Upper, Max=Maximum)**

| Context | Repository Overview | | | | Function Description | | | | Implementation Detail | | | | Testing | | | | Reconstruction Steps | | | |
|---|---|---|---|---|---|---|---|---|---|---|---|---|---|---|---|---|---|---|---|---|
| | Min | Low | Upp | Max | Min | Low | Upp | Max | Min | Low | Upp | Max | Min | Low | Upp | Max | Min | Low | Upp | Max |
| Repository Overview | 8 | 5 | 5 | 1 | - | - | - | - | - | - | - | - | - | - | - | - | - | - | - | - |
| Function Description | 2 | 0 | 0 | 0 | 0 | 1 | 0 | 0 | - | - | - | - | - | - | - | - | - | - | - | - |
| Implementation Detail | 24 | 17 | 17 | 31 | 0 | 0 | 0 | 0 | 2 | 5 | 3 | 5 | - | - | - | - | - | - | - | - |
| Testing | 52 | 62 | 60 | 54 | 0 | 0 | 0 | 0 | 3 | 3 | 6 | 1 | 1 | 1 | 1 | 1 | - | - | - | - |
| Reconstruction Steps | 3 | 2 | 0 | 2 | 0 | 0 | 0 | 0 | 1 | 0 | 3 | 1 | 0 | 0 | 0 | 0 | 0 | 0 | 1 | 0 |

**Table 2: Most Generated LoF Specification (Min=Minimum, Low=Lower, Upp=Upper, Max=Maximum)**

| Context | Repository Overview | | | | Function Description | | | | Implementation Detail | | | | Testing | | | | Reconstruction Steps | | | |
|---|---|---|---|---|---|---|---|---|---|---|---|---|---|---|---|---|---|---|---|---|
| | Min | Low | Upp | Max | Min | Low | Upp | Max | Min | Low | Upp | Max | Min | Low | Upp | Max | Min | Low | Upp | Max |
| Repository Overview | 1 | 0 | 0 | 0 | - | - | - | - | - | - | - | - | - | - | - | - | - | - | - | - |
| Function Description | 3 | 8 | 7 | 7 | 15 | 17 | 22 | 24 | - | - | - | - | - | - | - | - | - | - | - | - |
| Implementation Detail | 0 | 0 | 0 | 0 | 39 | 39 | 35 | 32 | 0 | 0 | 0 | 0 | - | - | - | - | - | - | - | - |
| Testing | 0 | 0 | 0 | 0 | 15 | 8 | 10 | 17 | 0 | 0 | 0 | 0 | 1 | 0 | 0 | 0 | - | - | - | - |
| Reconstruction Steps | 0 | 0 | 0 | 0 | 21 | 21 | 22 | 15 | 0 | 0 | 0 | 0 | 1 | 1 | 0 | 0 | 0 | 2 | 0 | 1 |

***Library Reconstruction Process***. To reconstruct the PyPI trivial libraries from our generated specifications, we analyzed 10 random libraries from each group (i.e., Minimum, Lower, Upper, Maximum groups), along with the most and least extensive generated specifications for each library. This resulted in 40 different libraries to be generated, each with two variants generated using the most and the least amount of specification content. We use the GitHub Copilot extension in Visual Studio Code and GPT-4.1 as the LLM. For the reconstruction prompt, we used only one type of prompt, since the sole context for library reconstruction was the generated AGENTS.md file. The prompt is below:

> **{Library Reconstruction Prompt}**
> *You are an autonomous software-development agent. You are tasked with generating a complete software library based strictly on a provided **AGENTS.md** specification file. This file describes the intended behavior, functions, interfaces, and usage of the library. Your goal is to fully recreate the library so that it behaves exactly as described in the specification.*
> *You must implement all functions, modules, and structures necessary for the library to operate as specified. Generate the full source code for the library, including all required files and folders. Output only code and file structure, with no additional explanations unless explicitly requested. You will be provided with a file named AGENTS.md that defines the library specification. Treat this file as the single source of truth. The library should contain the following contents:*
> *1. Project Setup*
> *2. Code Implementation*
> *3. Dependencies*
> *4. Tests*
> *5. Additional Details*
> *...*
> (Full prompt written in our repository)[3]

---

[3]anonymous.4open.science/r/acm-aiware-2026-reconstructing-trivial-pypi-packages

Upon reconstructing all libraries, we calculated the same metrics as before (LoF, CC, and FC). In addition, we counted the number of generated functions in both the original trivial libraries and the AI-generated libraries. Using these metrics, we compared the AI-generated libraries with the original human-written libraries. Through this comparison, we determined whether the AI agent introduced any differences, such as generating more or fewer LoF, CC, FC, and the number of functions. This analysis enables us to better understand how specification-driven generation influences the structural characteristics and complexity of the resulting libraries.

## 3 Results

We present our results below.

### 3.1 Deconstructing AGENTS.md

> **Findings 1**
>
> The context of *Function Description* and *Implementation Detail* generates the greatest amount of specification, while the context of *Repository Overview* and *Testing* generates the least amount of specification.

Table 1 and Table 2 show the context combinations that generate the least and the most amount of specification files, respectively. The combinations that share the same context represent single-context prompts, while the others represent paired-context prompts. We find that providing more context does not necessarily guarantee that GenAI will generate a more extensive specification. This is evident in the trend where the combination of Repository Overview and Testing—a paired-context prompt—produces the least amount of specification. This indicates that a single context can sometimes generate a greater amount of specification than a paired-context prompt. We also observe a clear trend that specifications containing the Function Description context generate the most extensive specifications. When paired with the Implementation Detail context, this

**Table 3: Least LoF Specification AI-generated Libraries Metrics Compared to Human-written Trivial Libraries**

| Samples | File Count (Δ) | | | Lines of File (Δ) | | | Cyclomatic Complexity (Δ) | | | Function Count (Δ) | | |
|---|---|---|---|---|---|---|---|---|---|---|---|---|
| | Min | Median | Max | Min | Median | Max | Min | Median | Max | Min | Median | Max |
| Minimum | -88.10% | -81.57% | -71.11% | -95.09% | -88.94% | -85.45% | -77.78% | -29.17% | +700.00% | -88.57% | -52.78% | +72.73% |
| Lower | -87.23% | -83.75% | -72.09% | -94.11% | -90.57% | -86.97% | -77.78% | -15.56% | +200.00% | -84.62% | -57.54% | +233.33% |
| Upper | -87.50% | -80.40% | -66.67% | -97.12% | -95.03% | -89.70% | -80.00% | -46.67% | +50.00% | -88.57% | -73.47% | +33.33% |
| Maximum | -94.23% | -86.89% | -76.53% | -99.98% | -98.36% | -96.44% | -100.00% | -33.33% | +66.67% | -92.49% | -83.71% | -6.67% |

**Table 4: Most LoF Specification AI-generated Libraries Metrics Compared to Human-written Trivial Libraries**

| Samples | File Count (Δ) | | | Lines of File (Δ) | | | Cyclomatic Complexity (Δ) | | | Function Count (Δ) | | |
|---|---|---|---|---|---|---|---|---|---|---|---|---|
| | Min | Median | Max | Min | Median | Max | Min | Median | Max | Min | Median | Max |
| Minimum | -84.62% | -80.78% | -74.47% | -86.91% | -83.02% | -80.20% | -50.00% | +12.50% | +400.00% | -74.29% | +14.55% | +82.35% |
| Lower | -88.46% | -80.01% | -65.12% | -93.00% | -87.40% | -80.56% | -50.00% | -5.56% | +100.00% | -43.75% | -23.63% | +120.00% |
| Upper | -83.67% | -81.25% | -55.36% | -94.54% | -89.81% | -82.47% | -62.50% | -5.00% | +33.33% | -63.83% | -31.34% | +200.00% |
| Maximum | -95.51% | -86.08% | -62.65% | -99.97% | -97.83% | -92.58% | -28.57% | 0.00% | +100.00% | -70.93% | -9.04% | +137.50% |

combination produces even more specification content compared to other context combinations.

## 3.2 Reconstructing the Library

> **Findings 2**
>
> AI-generated libraries tend to have *fewer LoF* and *lower FC*, regardless of the amount of specification provided. AI-generated libraries exhibit a higher maximum CC within a single function.

Table 3 presents the metrics of AI-generated libraries created using the least amount of specification, while Table 4 presents the metrics of libraries created using the most amount of specification. In both tables, the metrics are compared to the human-written trivial libraries. A red-labeled value indicates that the metric decreased, while a green-labeled value indicates that the metric increased. Across all cases, AI-generated libraries contain substantially fewer LoF and lower FC than the human-written libraries. This trend holds consistently across all four groups, all library samples, and for reconstructions using both the least and the most amount of specifications.

On the other hand, we observe that the maximum CC of a single function in the AI-generated libraries tends to increase compared to that of the human-written trivial libraries. In some cases, the CC increases to as much as seven times the original value found in the human-written libraries. Regarding function count, we observe that AI-generated libraries created with the least amount of specification generally produce fewer functions on average compared to human-written libraries. However, this trend does not hold for AI-generated libraries created with the most amount of specifications. In these cases, there are instances where the AI-generated libraries produce a higher function count—up to twice as many as the human-written libraries. On average, the difference between AI-generated libraries created with the most extensive specifications and human-written libraries remains relatively small compared to

the difference observed in AI-generated libraries created with the least amount of specification.

## 4 Research Agenda

With the growth of AI Slop [11] our key message is that specifications help guide the AI for more precise code for a specific functionality. Our results indicate that GenAI libraries generated from specifications are generally small and compact in size. We acknowledge potential limitations and threats to validity; therefore, a more comprehensive study is necessary to fully validate our findings. Future work includes developing a complete build pipeline that incorporates build processes, testing, deployment, and the potential execution of functions by client applications to enable the full realization of an on-demand library.

We envision two empirical lines of research for both the specifications and the on-demand GenAI libraries highlighted below.

**Specification Development.** The rise of AGENTS.md indicates that the future of agentic coding requires specific context to enable more streamlined and better correctness when generating code. The following research questions are needed:

- What is the best combination of specification contexts to generate correct libraries for a given situation?
- What level of human intervention is required when validating these specifications?
- How do these specifications vary according to the programming language or type of library?
- What metrics are appropriate for validating specifications?

**On-Demand Libraries.** Our results show that libraries generated by AI tend to be smaller in size. The following research questions therefore arise:

- What kinds of functions are more likely to be AI-generated, and which require more human intervention?
- What metrics are appropriate for measuring the quality of a GenAI library?
- What elements of a library cannot be replicated by an agent, and what constraints arise when generating a library?

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

Received 20 February 2007; revised 12 March 2009; accepted 5 June 2009

