# OpenReview forum: "AGENTS.md for Reconstructing a Tiny GenAI Library: Initial Analysis of Trivial PyPI Packages"
_ACM.org/AIWare/2026/Conference — Submitted to AIware 2026_

### Official Review · Reviewer_VeZK · 2026-03-10

**Rating:** 2
**Confidence:** 4

**Review:**

I think the author’s main question is well evaluated and the writing is easy to follow. However, two aspects of the paper are not suitably motivated, which I am hoping the authors can address:
- First, the approach uses natural language specifications to represent library functionality, but this choice is not justified: _Why is representing a library’s functionality as a natural language specification better than just representing them as code?_ Natural language specifications are a less precise and less efficient representation of functionality compared to the library’s actual implementation. If the goal is to generate a “custom” version of the library on the fly, one could just provide the original code directly to the LLM agent, which it could then adapt as it sees fit.
- Second, the need for generating libraries on the fly is not well motivated in the text. It sounds cool, but I’m not sure there is a strong need for it. _Why would we want to generate libraries on the fly? What is the benefit?_ Off the top of my head, I thought of two motivations:
    - Removing 3rd party dependencies: Taking on 3rd party dependencies in a software project is undesirable due to supply chain risks and CVEs. However, I don’t think the approach described actually removes 3rd party dependencies. We still must take a dependency on a 3rd party natural language specification which could still have CVEs and supply chain risks.
    - Functionality customization: this was briefly alluded to in the text, but there is no evidence that this might actually be useful in practice. Well-written libraries are intended to be generic, and generally should not need customization. _Could the authors provide citations or elaborate on this use case?_

My questions for the authors are italicized above.

**Summary:**

The authors study the feasibility of generating libraries on demand through natural language specifications of the library. They experiment with different prompting approaches for taking an existing library and converting it to a specification, and then they study the characteristics the reconstructed libraries. They focus on “trivial” PyPI packages for their evaluation. They report various metrics on the generated specifications and reconstructed libraries.

---

> ### Author Response · Authors · 2026-03-19
> **Rebuttal**
>
> Reviewer C (VeZK)
>
> Response to Q1: We represent library functionality in natural language because we envision a workflow where agents rely on context specification files (e.g., AGENTS.md [1, 2, 3]). We operate on the assumption that future agents will utilize these standardized files as primary grounding during code generation. Our assumption is not to work from code. We will add this in the limitations.
>
> Response to Q2: We appreciate the suggestion regarding dependency reduction. In the context of autonomous coding agents, it is a compelling research question whether an agent will choose to adopt an existing library or implement a custom solution on the fly. We will add this, along with the Reviewer’s points on CVEs and supply chain risks, as future research directions. Regarding customization and maintenance, while Mens and Decan [4] highlight the challenges of library maintenance, we argue that on-the-fly generation may offer a novel avenue for mitigating certain legacy risks.

---

### Official Review · Reviewer_6ZRX · 2026-03-11

**Rating:** 2
**Confidence:** 4

**Review:**

# Pros

- The paper explores a timely and relevant topic in agentic software engineering, focusing on the emerging role of AGENTS.md as a potential specification mechanism for AI agents.
- The experimental pipeline is clearly described, and the study is conducted at a relatively large scale for specification generation (5,760 AGENTS.md files).
- The proposed research agenda identifies several open questions that could inspire future work in specification engineering for AI agents.

# Cons

- The main finding on specification length is highly intuitive and offers limited insight. The paper reports that Function Description and Implementation Detail contexts generate the greatest amount of specification, while Repository Overview and Testing generate the least.
This result is largely expected and aligns with common intuition about documentation granularity. As such, it provides limited new insight into how agent-oriented specifications differ from conventional documentation practices.


- A central empirical result of the paper is that AI-generated libraries consistently exhibit fewer lines of code and fewer files, regardless of the amount of specification provided, while showing an increased maximum cyclomatic complexity within a single function.
While this observation is interesting, it significantly weakens the paper’s core motivation. If richer or more detailed specifications do not materially influence the overall structure of the generated libraries, it remains unclear what role specification design actually plays in guiding agentic code generation.

**Summary:**

This paper presents an initial empirical study on using AGENTS.md as a specification artifact to guide the reconstruction of trivial PyPI libraries by generative AI agents.
The authors generate multiple AGENTS.md specifications using different combinations of context types (e.g., Repository Overview, Function Description, Implementation Detail, Testing), and then use these specifications to reconstruct libraries with an AI agent. The study analyzes both the generated specifications and the reconstructed libraries using structural metrics such as Lines of File (LoF), File Count (FC), Cyclomatic Complexity (CC), and Function Count.
The paper aims to explore how different specification contexts affect (1) the amount of generated specification content and (2) the structural characteristics of AI-generated libraries, and positions itself as a new ideas paper proposing a research agenda for specification-driven, on-demand GenAI libraries.

---

> ### Author Response · Authors · 2026-03-19
> **Rebuttal**
>
> Reviewer B (6ZRX)
>
> Response to Q1: Similar to our response to Reviewer A (Q2), our investigation into specification length is driven by the practical constraints of context windows and token efficiency. While this aligns with common intuition, our goal was to empirically validate these assumptions and use these combinations as a baseline for automated library generation. To clarify, our research is not about whether human written specifications are better than GenAI.
>
> Response to Q2: We agree that the precise role of "specification design" in guiding agentic code generation warrants further investigation. While we have not yet executed functional testing on the generated code, our findings suggest that specifications significantly reduce code verbosity. Our immediate next phase of research will focus on identifying which specific contexts yield functionally correct code which should provide insights on the cost performance. We will add this for future work.

---

### Official Review · Reviewer_dH9N · 2026-03-11

**Rating:** 3
**Confidence:** 3

**Review:**

Positives
+ This paper addresses an emerging and interesting problem, i.e., which contexts in AGENTS.md contribute most to library reconstruction.
+ The paper conducts systematic data collection and experiments.

Negatives
- The practical significance of Finding 1 is unclear.

Questions

1. The abstract states that “specifications have potential to guide GenAI for tiny code compared to human-written equivalents” (line 20). However, the study mainly compares structural metrics (e.g., LoF and FC) of the generated libraries, while the correctness of the generated code is not evaluated. Therefore, this claim may be somewhat strong given the current evaluation.

2. The paper shows that some context combinations generate longer or shorter specifications (Finding 1), where specification size is measured by Lines of File (line 226). However, the practical significance of this finding is unclear, as the paper does not explain how specification length relates to the quality or correctness of the reconstructed libraries. Could the authors clarify the motivation for analyzing specification length and whether it has any impact on the generated libraries?

3. In Section 3.2, the paper only considers up to two-context combinations when generating specifications for library reconstruction, and concludes that AI-generated libraries tend to have fewer LoF and lower FC. However, this design may limit the conclusions, as other combinations of contexts (e.g., using all five contexts to generate specifications) could potentially lead to different reconstruction results. It would be helpful if the authors could discuss whether richer specifications (e.g., combining all contexts) would change the observed trends.

**Summary:**

This paper explores generating libraries on-demand by decomposing specification contexts in AGENTS.md. The authors generate specifications using different context combinations, reconstruct libraries based on them, and compare the generated libraries with human-written equivalents. They find that Function Description and Implementation Detail produce longer specifications, while Repository Overview and Testing produce shorter ones. They also observe that AI-generated libraries tend to have fewer LoF and lower FC, but higher maximum CC within a single function.

---

> ### Author Response · Authors · 2026-03-19
> **Rebuttal**
>
> Reviewer A (dH9N)
>
> Response to Q1: We acknowledge that our initial claims were overstated. We will revise the title and introduction to better reflect the current scope. Additionally, we will explicitly categorize the evaluation as immediate future work in the concluding sections.
>
> Response to Q2: Regarding the significance of Finding 1, prior work [1, 2, 5] suggests that AGENTS.md is beneficial for coding agents, though the specific extent remains unquantified. Given that token usage is the primary "currency" for models like GPT-4, Claude, and Grok, understanding how different contexts impact the window is critical. This motivated our analysis of specification length, and we will clarify this connection. We will add this discussion of token usage in the introduction and future work.
>
> Response to Q3: We initially tested all combinations (31 combinations) but omitted several technical results due to page limits and complexity of the results. As this is a vision paper intended to demonstrate potential, we focused on core findings. We intend to provide the comprehensive results of all combinations in our upcoming work.

---

> > ### Comment · Reviewer_dH9N · 2026-03-20
> >
> > Thank you for the clarifications. The responses address my questions and help improve the clarity of the paper.

---

### Author Response · Authors · 2026-03-19
**Rebuttal**

We thank the reviewers for their insightful comments. We will incorporate all suggested references and clarifications into the camera-ready version of the manuscript.

Reviewer A (dH9N)

Response to Q1: We acknowledge that our initial claims were overstated. We will revise the title and introduction to better reflect the current scope. Additionally, we will explicitly categorize the evaluation as immediate future work in the concluding sections.

Response to Q2: Regarding the significance of Finding 1, prior work [1, 2, 5] suggests that AGENTS.md is beneficial for coding agents, though the specific extent remains unquantified. Given that token usage is the primary "currency" for models like GPT-4, Claude, and Grok, understanding how different contexts impact the window is critical. This motivated our analysis of specification length, and we will clarify this connection. We will add this discussion of token usage in the introduction and future work.

Response to Q3: We initially tested all combinations (31 combinations) but omitted several technical results due to page limits and complexity of the results. As this is a vision paper intended to demonstrate potential, we focused on core findings. We intend to provide the comprehensive results of all combinations in our upcoming work.

Reviewer B (6ZRX)

Response to Q1: Similar to our response to Reviewer A (Q2), our investigation into specification length is driven by the practical constraints of context windows and token efficiency. While this aligns with common intuition, our goal was to empirically validate these assumptions and use these combinations as a baseline for automated library generation. To clarify, our research is not about whether human written specifications are better than GenAI.

Response to Q2: We agree that the precise role of "specification design" in guiding agentic code generation warrants further investigation. While we have not yet executed functional testing on the generated code, our findings suggest that specifications significantly reduce code verbosity. Our immediate next phase of research will focus on identifying which specific contexts yield functionally correct code which should provide insights on the cost performance. We will add this for future work.

Reviewer C (VeZK)

Response to Q1: We represent library functionality in natural language because we envision a workflow where agents rely on context specification files (e.g., AGENTS.md [1, 2, 3]). We operate on the assumption that future agents will utilize these standardized files as primary grounding during code generation. Our assumption is not to work from code. We will add this in the limitations.

Response to Q2: We appreciate the suggestion regarding dependency reduction. In the context of autonomous coding agents, it is a compelling research question whether an agent will choose to adopt an existing library or implement a custom solution on the fly. We will add this, along with the Reviewer’s points on CVEs and supply chain risks, as future research directions. Regarding customization and maintenance, while Mens and Decan [4] highlight the challenges of library maintenance, we argue that on-the-fly generation may offer a novel avenue for mitigating certain legacy risks.

[1] Gloaguen, T., Mündler, N., Müller, M., Raychev, V., & Vechev, M. (2026). Evaluating AGENTS. md: Are Repository-Level Context Files Helpful for Coding Agents?. arXiv preprint arXiv:2602.11988.

[2] Chatlatanagulchai, W., Thonglek, K., Reid, B., Kashiwa, Y., Leelaprute, P., Rungsawang, A., ... & Iida, H. (2025, November). On the use of agentic coding manifests: An empirical study of claude code. In International Conference on Product-Focused Software Process Improvement (pp. 543-551). Cham: Springer Nature Switzerland.

[3] Lulla, J. L., Mohsenimofidi, S., Galster, M., Zhang, J. M., Baltes, S., & Treude, C. (2026). On the Impact of AGENTS. md Files on the Efficiency of AI Coding Agents. arXiv preprint arXiv:2601.20404.

[4] Mens, T., & Decan, A. (2024). An overview and catalogue of dependency challenges in open source software package registries. arXiv preprint arXiv:2409.18884.

[5] Galster, M., Treude, C., & Baltes, S. (2025). Context engineering for AI agents in open-source software. arXiv preprint arxiv:2510.21413